# Glucose Supplementation Enhances the Bactericidal Effect of Penicillin and Gentamicin on *Streptococcus sanguinis* Persisters

**DOI:** 10.3390/antibiotics14010036

**Published:** 2025-01-05

**Authors:** Kazuya Takada, Yoshie Yoshioka, Kazumasa Morikawa, Wataru Ariyoshi, Ryota Yamasaki

**Affiliations:** 1Division of Infections and Molecular Biology, Department of Health Promotion, Kyushu Dental University, Kitakyushu 803-8580, Fukuoka, Japan; r20takada@fa.kyu-dent.ac.jp (K.T.); r16yoshioka@fa.kyu-dent.ac.jp (Y.Y.); arikichi@kyu-dent.ac.jp (W.A.); 2Division of Oral Functional Development, Department of Health Promotion, Kyushu Dental University, Kitakyushu 803-8580, Fukuoka, Japan; r24morikawa@fa.kyu-dent.ac.jp; 3Collaborative Research Centre for Green Materials on Environmental Technology, Kyushu Institute of Technology, 1-1 Sensui-chou, Tobata-ku, Kitakyushu 803-8580, Fukuoka, Japan

**Keywords:** *Streptococcus sanguinis*, persister, infective endocarditis

## Abstract

**Background**: *Streptococcus sanguinis* is a leading cause of infective endocarditis (IE), which causes diverse clinical symptoms and even death. Recurrence after treatment is a crucial problem in IE, possibly caused by the presence of “persister” cells, a small bacterial population that can survive antimicrobials. In this study, the residual risk for penicillin G (PCG) and gentamicin (GM), used for treating IE, to induce *Streptococcus sanguinis* persisters, was investigated. **Methods**: The bactericidal effects of PCG and GM on *S. sanguinis* were evaluated. Furthermore, we confirmed whether the *S. sanguinis* that survived following combination treatment with PCG and GM were persisters. The bactericidal effect of the combination of PCG and GM against *S. sanguinis* was measured after the addition of glucose or arginine. **Results**: Following 48 h of treatment with PCG (1600 μg/mL) and GM (64 μg/mL), *S. sanguinis* survived, albeit with a low bacterial count, indicating the presence of persisters. The addition of glucose or arginine to PCG and GM increased the bactericidal effect on residual persister cells and reduced the number of persister cells. Moreover, the addition of glucose at concentrations of 10 mg/mL or higher was substantially effective in achieving sterilization. **Conclusions**: Our findings demonstrate that persisters of *S. sanguinis* that survive antimicrobial treatment may make the treatment of IE challenging, and that combining antimicrobial treatment with glucose is effective for eliminating persisters of *S. sanguinis*. Taken together, these findings may facilitate the development of novel therapeutic strategies against IE caused by oral streptococcal infection.

## 1. Introduction

Oral diseases (dental caries and periodontal disease) [1] are primarily caused by oral bacteria and are widely prevalent worldwide. Oral bacteria can cause not only oral, but also systemic diseases, such as infective endocarditis (IE) [2]. *Streptococcus sanguinis* is widely present in the oral cavity from early infancy and is found on tooth surfaces and supragingival plaques [3,4]. It is usually associated with healthy plaque biofilms [5] and contributes to the subsequent microbial colonization that is essential in biofilm formation [6]. In addition to its role as a primary oral colonizer, *S. sanguinis* is a well-known cause of IE [7]. IE is a systemic septic disease that forms bacterial plaques in the valve leaflets and endocardium. It presents with various clinical manifestations, including bacteremia, vascular emboli, and cardiac injury [8]. IE is caused by various bacteria, including *Staphylococcus aureus* and viridans group streptococci (including *S. sanguinis*). Its annual incidence is reported to be rare, ranging from three to ten cases per 100,000 people [8]. However, once IE occurs, it is associated with an elevated mortality rate [9]. *S. sanguinis* causes IE by invading the body through capillaries in the gingiva during dental treatments and brushing [10]. IE is typically initially treated with antimicrobial agents [11]; however, recurrence after treatment is common. Specifically, the reported recurrence rate is approximately 6%, which is not a low rate [12]. IE recurrence is caused by the regrowth of infected microorganisms, highlighting that conventional antimicrobial therapy does not fully eliminate infectious microorganisms. One possible reason for this is the presence of “persisters,” which we consider to be a key risk factor for IE recurrence.

Hobby et al. first observed persisters in 1942 [13]. Persisters are bacterial phenotypes formed as a result of environmental stresses, such as drugs or starvation [14,15]. Their major difference from resistant bacteria is that the persisters that survive stress do not acquire resistance genes, i.e., the persister cells and the other sensitive cells remain in the same genome. Persister against antibiotics is defined as the ability of susceptible bacteria to survive the antibiotics by entering a nongrowing state [16]. A key characteristic of persisters against antibiotics is two-phase sterilization, in which susceptible populations are rapidly sterilized, while subpopulations of persisters are sterilized after a delay [17]. This phenomenon is indicative of phenotypic heterogeneity in bacterial cultures, meaning that not all bacteria in a population are killed at the same rate. Unlike resistant bacteria, persisters do not grow in the presence of antibiotics. Both persister and antibiotic-resistant bacteria exhibit increased survival rates in the presence of antibiotics compared to susceptible bacteria. However, in the presence of antibiotics, only a small fraction of bacteria survive as persisters, whereas the entire population of resistant bacteria survives [18]. Therefore, antibiotic persisters represent a bacterial phenotype that allows a very small population of bacteria to survive antibiotic exposure, and they are distinct from drug-resistant bacteria. Since the discovery of this population, the underlying mechanisms have been elucidated. The toxin/antitoxin system was first shown to be important as a survival mechanism for persisters [19], and many TA systems related to persisters have been identified. While extensive research on persisters has focused on common environmental bacteria, such as *Escherichia coli*, *Staphylococcus aureus*, and *Pseudomonas aeruginosa* [20,21,22,23], persisters have also been reported for oral bacteria [24,25,26]. As mentioned above, oral streptococci are closely related to IE, but there have been no reports of oral streptococcal persisters being involved, to the best of our knowledge. Even with IE antimicrobial therapy, persisters of oral-associated bacteria can survive and proliferate, causing chronic infection and recurrence.

Penicillin G (PCG), gentamicin (GM), and vancomycin are used as treatments for IE, based on international guidelines established for prescribing medications (such as The Japanese Circulation Society [27], the American Heart Association (AHA) [28], and the European Society of Cardiology (ESC) [11]). However, the role of bacterial persisters in the recurrence and chronicity of IE has not been reported in previous studies or guidelines. Therefore, this study aimed to investigate whether *S. sanguinis* survives by forming persisters following treatment with PCG and GM, as well as to identify effective treatment strategies against persisters. Identifying survival through persister formation and developing methods to effectively sterilize persisters will provide new strategies to treat infections in the future.

## 2. Results

### 2.1. Minimum Inhibitory Concentration (MIC) of PCG and GM Against S. sanguinis

The growth inhibition of *S. sanguinis* by PCG and GM is shown in Figure 1.

The growth inhibition of PCG and GM against *S. sanguinis* was evaluated by measuring absorbance at 620 nm, and then compared to an untreated control (0 µg/mL PCG and GM). PCG and GM significantly inhibited the growth of *S. sanguinis* at concentrations of 0.06 μg/mL and 64 μg/mL, respectively; therefore, these were determined as the MICs.

### 2.2. S. sanguinis Continues to Survive by Forming Persisters Against PCG and GM

The bactericidal effects against *S. sanguinis* were evaluated to measure the colony-forming units (CFU) using PCG at 1600 μg/mL and GM at 64 μg/mL, which are the concentrations specified in the AHA guidelines. The combination of PCG and GM (PCG/GM) was highly effective, killing 99.8% of the *S. sanguinis* population in 24 h. However, some of the bacteria continued to survive after that time (Figure 2).

The surviving bacteria likely escaped the antibacterial effects and were considered persister cells. The bactericidal effect of the individual PCG or GM treatments was lower than that of the combined treatment (Appendix A).

### 2.3. S. sanguinis That Survived PCG/GM Is Not Drug-Resistant But Persister

*S. sanguinis* that survived PCG/GM treatment for more than 24 h may have acquired resistance genes and survived; therefore, we determined whether the surviving bacteria were persisters or drug-resistant. As shown in Figure 3, a second round of PCG/GM treatment on the bacterial population that had regrown after an initial PCG/GM treatment reduced the bacterial count to the same extent.

The number of drug-resistant mutants did not decrease (i.e., they did not die) when retreated with the antibiotics again, indicating that the bacteria surviving the PCG/GM treatment were persisters. This result highlights the risk of recurrence and the challenges in achieving effective treatment using current drug treatments, as persisters remain regardless of repeated antibiotic treatment.

### 2.4. Glucose Enhances the Bactericidal Effect of PCG/GM Against S. sanguinis Persisters

Since a small percentage of bacteria survived by becoming persisters following treatment with PCG/GM, we subsequently explored the sterilization of these persister cells. A combination of glucose and aminoglycoside antimicrobials has been demonstrated to be effective in sterilizing persister cells [29]. Amino acids can resuscitate persister cells from dormancy [22]. We hypothesized that awakening the persister cells from their dormant state would enhance the bactericidal effect of the antimicrobials. Arginine was used in this experiment because arginine can be metabolized by *S. sanguinis* [30]. Consequently, we evaluated whether the addition of glucose (1, 10, or 100 mg/mL) or arginine (725 μg/mL or 10 mg/mL) to an M9 minimal medium containing PCG/GM enhanced the bactericidal effect of the antimicrobials. As shown in Figure 4a, all glucose concentrations resulted in enhanced bactericidal activity. In addition, 10 and 100 mg/mL of glucose completely killed the cells after 24 h of treatment. As shown in Figure 4b, 10 mg/mL of arginine resulted in enhanced bactericidal activity.

However, at a glucose concentration of 1 mg/mL, which is similar to the average concentration in human blood [31], the number of viable cells was reduced compared to that in the group without glucose, although only a few cells survived. A few cells survived the addition of 10 mg/mL arginine as well. Next, 10 mg/mL of glucose or arginine was added to the *S. sanguinis* persisters that formed after the treatment with PCG/GM, and these cells were completely eliminated after 3 h (glucose) and 24 h (arginine) (Figure 5).

Collectively, these results demonstrated the effectiveness of combining glucose or arginine with antibiotics in sterilizing persisters. Furthermore, glucose alone (without antibiotics) did not exhibit bactericidal effects (Appendix A), indicating that glucose does not play a direct role in bactericidal activity, but enhances the antimicrobial effects of the drugs on persister cells.

## 3. Discussion

### 3.1. Persisters Play a Crucial Role in the Treatment of IE

This study investigated the role of causative bacterial persisters in the antimicrobial treatment of IE, and the risk of recurrence. Specifically, we investigated *S. sanguinis*, a typical causative agent of IE, and used PCG and GM, which are typical therapeutic agents indicated in treatment guidelines. Although they are not used clinically, glucose or arginine were used with PCG and GM. Glucose in combination may enhance the uptake of an antimicrobial by increasing the proton motive force (PMF) [29]. Arginine may resuscitate persister cells from their dormant state, making them unable to evade the effects of antimicrobial agents [22]. Generally, antibiotics eliminate bacteria in blood vessels, effectively treating the disease; however, there is concern that the presence of persisters, which partially evade the action of antibiotics, can lead to recurrence. Therefore, this study hypothesized that sterilization of the persisters, along with the complete removal of the causative organisms, can cure the infection and reduce the risk of chronicity and recurrence. Previous studies have shown that PCG/GM promotes bactericidal effects through synergistic mechanisms [32]. In this study, PCG/GM was more effective in killing bacteria than either treatment alone. Therefore, the value of combination therapy using the drugs indicated in the guidelines was also demonstrated in this study. However, there was a limited bactericidal effect after 24 h, and a biphasic bactericidal curve was observed, indicating the presence of viable bacteria despite the synergistic effects of the PCG/GM combination treatment (Figure 2). Correspondingly, this biphasic bactericidal curve and the absence of drug-resistance genes (Figure 3) were consistent with previously reported characteristics of persisters [17]. Therefore, this study suggests that persisters may form during the treatment of IE, and that, together with previously reported resistant bacteria [33,34], persisters may make the treatment of infections more challenging.

Next, we examined the elimination of persisters. Previous reports have shown that anticancer agents such as cisplatin and mitomycin C are effective in killing *E. coli* persisters [35,36]. However, there are concerns about the high burden of these drugs on the human body due to their potent effects. In this study, we sought a substance with a low impact on the human body and, thus, focused on glucose. Glucose, in combination with aminoglycoside antimicrobials, can enhance the bactericidal effect on persister cells by increasing the PMF and uptake of antimicrobial agents. Moreover, a previous study demonstrated that treatment with glucose and GM effectively kills *Escherichia coli* persisters [29]. Similar results were obtained in the present study with the addition of glucose, which enhanced the bactericidal effect of the antibiotics against *S. sanguinis*. In particular, the addition of glucose at concentrations above 10 mg/mL resulted in complete sterilization and was extremely effective (Figure 4 and Figure 5). Though the complete mechanism of aminoglycoside uptake is unclear, it is known that a threshold PMF is required. Glucose is transported to the cytoplasm and enters glycolysis, where its catabolism generates nicotinamide adenine dinucleotide. This compound is oxidized by enzymes in the electron transport chain, contributing to PMF. We considered that elevated PMF enhances aminoglycoside uptake and the bactericidal effect [29]. Therefore, the addition of glucose to antibiotic therapy may be an effective strategy for eliminating *S. sanguinis* persisters formed by antimicrobial agents. A previous study demonstrated that amino acids resuscitate *E. coli* persister cells from their dormant state [22]. Persister cells resuscitated from dormancy cannot evade the bactericidal effects of antimicrobials, as well as susceptible bacteria. Arginine, an amino acid, is used in the oral cavity primarily to prevent dental caries. *S. sanguinis* metabolizes arginine through the arginine deiminase pathway to produce citrulline, ornithine, CO_2_, adenosine triphosphate, and ammonia [30]. This metabolism neutralizes the acidic environment and reduces tooth demineralization. Arginine metabolism could be a trigger to resuscitate the persister state, and the effect of arginine on the bactericidal effect was examined. Addition of 10 mg/mL arginine enhanced the bactericidal effect of the antibiotics against *S. sanguinis* (Figure 4 and Figure 5). The addition of arginine may also be an effective strategy for eliminating *S. sanguinis* persister cells formed by antimicrobial agents. However, the enhancement of the bactericidal effect was lower when compared to the same concentration of glucose. Therefore, we believe that adding glucose is a more effective strategy.

### 3.2. Study Limitations

In this study, we focused on *S. sanguinis* for the treatment of IE caused by oral bacteria. However, IE is a complex biofilm infection caused by various bacterial species, mainly occurring in the endocardium. Further studies are required to investigate complex biofilms with various bacterial species in antimicrobial therapy in clinical settings. In addition, we found that the addition of glucose at concentrations of 10 mg/mL or higher can enhance the effectiveness of antibiotics and eradicate bacterial populations, including persisters; however, the safety of glucose doses above 10 mg/mL in the human body should be examined, as these doses are much higher than normal blood glucose levels [37], and the treatment lasts 2 to 6 weeks [27]. The maintenance of excessively high blood glucose levels may cause diabetes [38]. In the future, it will be necessary to further examine patients before and after antibiotic treatment to clinically determine the presence of bacteria that escape drug treatment. Furthermore, the details on how glucose works with antibiotics to kill persisters need to be clarified. *S. sanguinis* can also metabolize other sugars, such as sucrose [39]. Therefore, it is important to verify whether the effectiveness of drug treatments, such as the combination with glucose used in this study (or other carbon sources), can be further enhanced.

## 4. Materials and Methods

### 4.1. Microbial Strains and Culture Medium

*S. sanguinis* ATCC10556 is held by the American Type Culture Collection (ATCC; Manassas, VA, USA) as a strain isolated from a patient with subacute bacterial endocarditis [40]. This strain was obtained from ATCC and used as the target bacterium. *S. sanguinis* was seeded from the glycerol stock onto brain–heart infusion (BD, Franklin Lakes, NJ, USA) agar plates containing 1% yeast extract (BD) (BHIY), at 37 °C, and incubated for 1 d at 37 °C with 5% CO_2_. Single colonies were seeded in BHIY broth and incubated at 37 °C in 5% CO_2_ for 15 h. BHIY and M9 minimal medium (Na_2_HPO_4_·12H_2_O 15.13 g/L, KH_2_PO_4_ 3 g/L, NaCl 0.5 g/L, NH_4_Cl 0.5 g/L, MgSO_4_·7H_2_O 246.5 mg/L, and CaCl_2_·2H_2_O 14.7 mg/L) were used in the experiment. Na_2_HPO_4_·12H_2_O was obtained from Kishida Chemical Co., Ltd. (Osaka, Japan). KH_2_PO_4_, NaCl, NH_4_Cl, MgSO_4_·7H_2_O, and CaCl_2_·2H_2_O were obtained from FUJIFILM Wako Pure Chemical Co., Ltd. (Osaka, Japan).

### 4.2. MIC Measurement

The microliquid dilution method was used to determine the MIC of *S. sanguinis* against PCG or GM. BHIY was prepared with 4 μg/mL of PCG (Tokyo Chemical Industry Co., Ltd., Tokyo, Japan) and 256 μg/mL of GM (Sigma-Aldrich, St. Louis, MO, USA), and it was serially diluted twofold on 96-well plates (Iwaki, AGC Techno Glass Co., Ltd., Shizuoka, Japan). Each well was seeded to achieve an absorbance of 0.05 at 600 nm, and then incubated at 37 °C with 5% CO_2_ for 24 h. Growth inhibition was measured at 620 nm using a microplate reader (Multiskan FC, Thermo Fisher Scientific, Waltham, MA, USA), and the concentration at which growth was significantly inhibited was defined as the MIC. Controls were set at 0 μg/mL. All experiments were conducted with at least three biological replicates.

### 4.3. Bactericidal Effect of PCG and GM Against S. sanguinis

Overnight cultures of *S. sanguinis* were reinoculated in fresh BHIY broth and incubated at 37 °C in 5% CO_2_ until an absorbance of 0.4 at 600 nm was reached. The bacterial pellet was collected by centrifugation at 4000× *g* for 10 min. The PCG and GM concentrations were determined based on the IE guidelines published by the American Heart Association [28], and on human blood volume [41]. The cells were resuspended in BHIY containing PCG (1600 μg/mL)/GM (64 μg/mL) and incubated for 6, 18, 24, 36, and 48 h. After each incubation, the bacteria were collected, washed twice with PBS, serially diluted tenfold, spotted on BHIY agar plates, and the CFU were measured. Controls were cultured with BHIY only.

### 4.4. Confirmation of Persistence

*S. sanguinis* reinoculated culture (OD600 = 0.4) was centrifuged at 4000× *g* for 10 min. After suspension in BHIY containing PCG (1600 μg/mL)/GM (64 μg/mL), the cells were incubated for 24 h at 37 °C with 5% CO_2_. Centrifugation was performed at 4000× *g* for 10 min, and bacterial pellets were collected and washed twice with PBS. Fresh BHIY was added and incubated in an antibiotic-free environment for 24 h (37 °C, 5% CO_2_). The regrown culture was centrifuged at 4000× *g* for 10 min, and the bacterial pellet was collected and resuspended in BHIY containing the same concentration of PCG/GM. Following this, the suspension was incubated at 37 °C with 5% CO_2_ for 24 h. After each experiment, the bacteria were collected, washed twice with PBS, serially diluted tenfold, spotted on BHIY agar plates, and the CFU were measured.

### 4.5. Effect of Glucose and Arginine on S. sanguinis Persisters

Initially, an experiment in which antibiotics and glucose were simultaneously applied was conducted. *S. sanguinis* reinoculated culture (OD600 = 0.4) was centrifuged at 4000× *g* for 10 min. The bacterial pellets were resuspended in M9 minimal medium containing PCG (1600 μg/mL)/GM (64 μg/mL) and 1, 10, or 100 mg/mL of glucose (Tokyo Chemical Industry Co., Ltd., Tokyo, Japan) or arginine (725 μg/mL or 10 mg/mL) (FUJIFILM Wako Pure Chemical Co., Ltd., Osaka, Japan), and incubated for 6, 18, 24, 36, and 48 h (37 °C, 5% CO_2_). To accurately determine the effects of glucose and arginine, M9 was used instead of BHIY medium, which is less affected. After each incubation period, the cells were washed twice with PBS, and tenfold serial dilutions were prepared and spotted on BHIY agar plates to measure the CFU. Following this, an experiment was conducted, in which glucose or arginine was applied to the persisters that survived the antibiotic treatment. *S. sanguinis* cultured under the same conditions was treated with M9 minimal medium containing PCG/GM for 24 h and centrifuged at 4000× *g* for 10 min to collect the persister cells. The cells were washed twice with PBS and treated with M9 minimal medium containing PCG/GM and glucose or arginine (10 mg/mL). The cells were then incubated for 1, 3, 6, 18, or 24 h. After each incubation period, the cells were washed twice with PBS, tenfold serial dilutions were prepared, spotted on BHIY agar plates, and the CFU were measured. In both experiments, controls were treated with M9 minimal medium containing PCG/GM.

### 4.6. Statistical Analysis

The statistical tests and sample details are shown in the figure legends. All experiments were repeated at least three times. Error bars indicate the standard deviation from the mean. Microsoft Excel was used to organize raw data and to generate figures. EZR was used for statistical analysis to check the normal distribution of the samples and equal variances, as well as to perform ANOVA and multiple comparisons.

## 5. Conclusions

This study highlights how persisters may serve as a risk factor that prevents effective antibiotic treatment against *S. sanguinis*, which is a causative organism of IE. Notably, we revealed that the presence of *S. sanguinis* persisters prevented complete eradication by antibiotics, while the combination of antimicrobials and glucose effectively eradicated persisters. The *S. sanguinis* persisters identified in this study suggest that they may cause chronicity and recurrence in IE. Taken together, these findings may guide the development of new treatment strategies for IE caused by oral streptococcal infections.

## Figures and Tables

**Figure 1 antibiotics-14-00036-f001:**
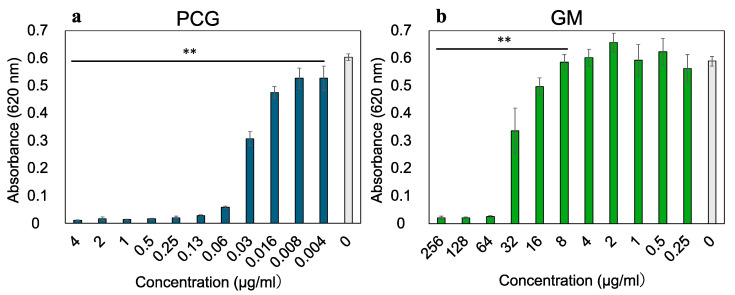
Minimum inhibitory concentration (MIC) of penicillin G (PCG) (**a**) and gentamicin (GM) (**b**) against *S. sanguinis* in brain–heart infusion broth with yeast extract (BHIY). BHIY broth included different concentrations of PCG and GM, which were incubated for 24 h, and the absorbance was measured at 620 nm. Error bars indicate the standard deviation from at least three biological replicates. ANOVA with Tukey’s multiple comparisons was used to compare the controls with the other groups. (** *p* < 0.01).

**Figure 2 antibiotics-14-00036-f002:**
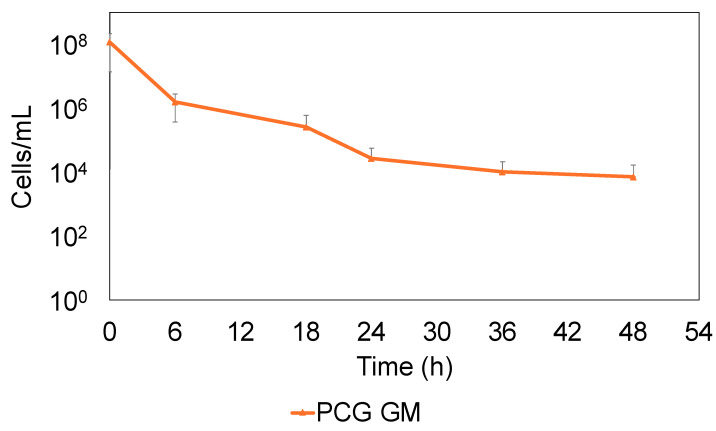
Bactericidal effects of combined PCG (1600 µg/mL)/GM (64 µg/mL) treatment against *S. sanguinis* in the exponential phase. CFU were measured at 0, 6, 18, 24, 36, and 48 h. Error bars indicate the standard deviation across at least three biological replicates (the experiments were performed in cultures from different single colonies).

**Figure 3 antibiotics-14-00036-f003:**
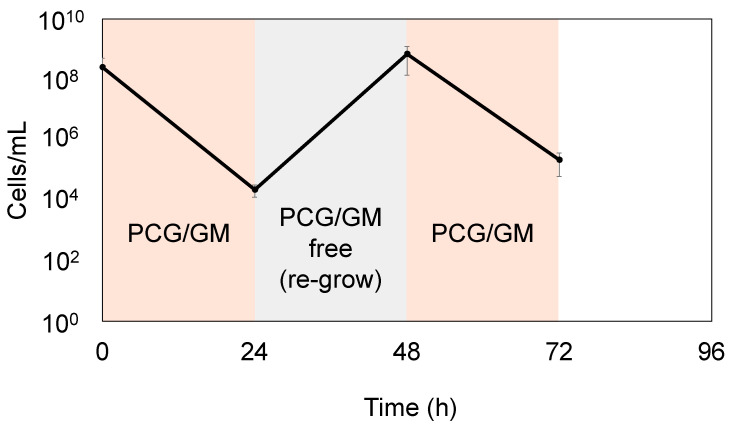
Persister formation in *S. sanguinis* following PCG/GM treatment and recovery in BHIY medium. After treatment with PCG (1600 μg/mL)/GM (64 μg/mL) for 24 h, the medium was changed to BHIY only, and cells were incubated at 37 °C in 5% CO_2_ for 24 h. The regrown *S. sanguinis* was retreated with the same concentration of PCG/GM for 24 h. Error bars indicate the standard deviation across at least three biological replicates.

**Figure 4 antibiotics-14-00036-f004:**
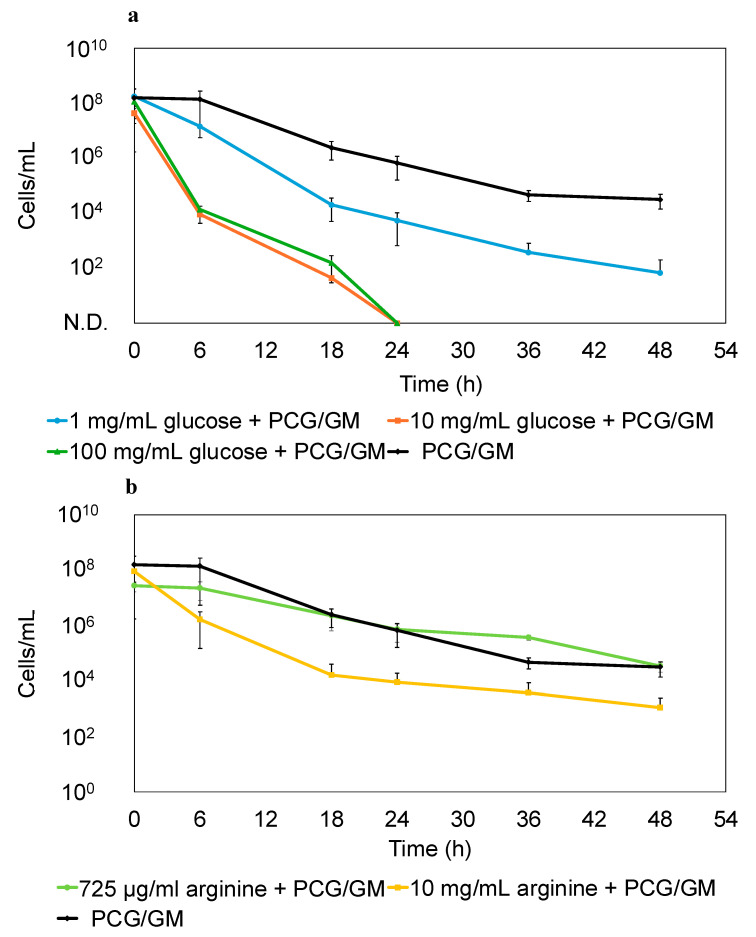
Bactericidal effect of PCG (1600 μg/mL)/GM (64 μg/mL) on *S. sanguinis* in M9 minimal medium following the addition of glucose (1, 10, or 100 mg/mL) (**a**) and arginine (725 μg/mL or 10 mg/mL) (**b**). The CFU were measured at 0, 6, 18, 24, 36, and 48 h. Error bars indicate the standard deviation from at least three biological replicates. N.D. indicates not detected.

**Figure 5 antibiotics-14-00036-f005:**
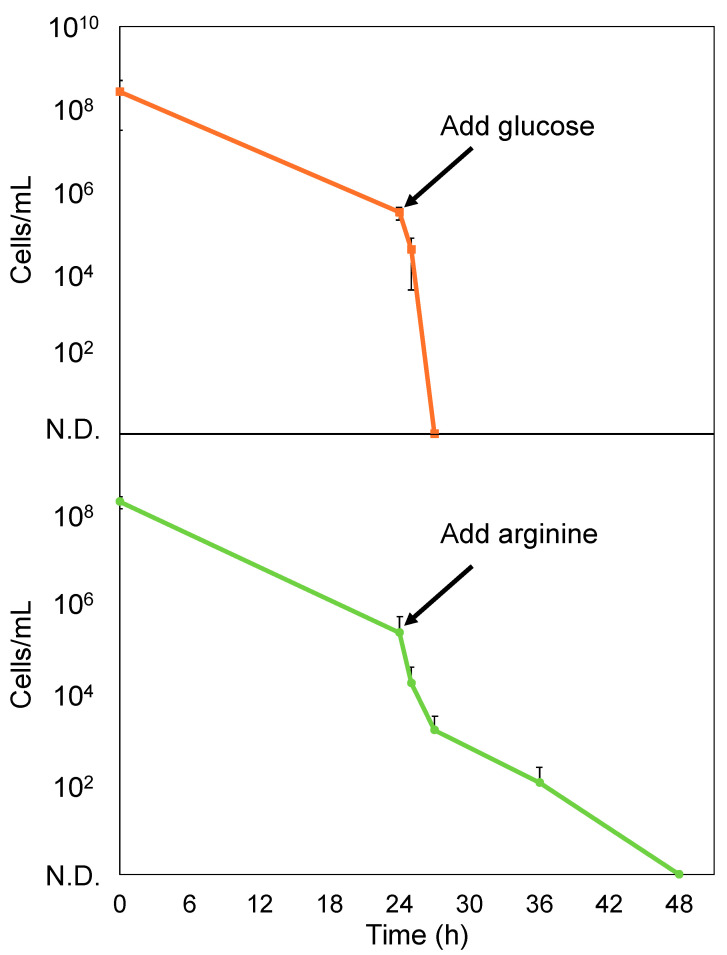
Bactericidal effects of glucose and arginine on *S. sanguinis* persister cells after PCG/GM treatment. Cells were incubated with M9 minimal medium and treated with PCG (1600 μg/mL)/GM (64 μg/mL) for 24 h. Following this, the cells were washed with PBS and retreated with the same concentration of PCG/GM in M9 minimal medium containing 10 mg/mL of glucose or arginine. Error bars indicate the standard deviation from at least three biological replicates. N.D. indicates not detected.

## Data Availability

The data presented in this article are available upon request from the corresponding author.

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
