# Peer review of "Glucose Supplementation Enhances the Bactericidal Effect of Penicillin and Gentamicin on Streptococcus sanguinis Persisters"

_antibiotics, 2025, doi:10.3390/antibiotics14010036_

Round 1
Reviewer 1 Report
Comments and Suggestions for Authors
The article aims to investigate whether the combination therapy of PCG and GM leads to the emergence of persister cells of S. sanguinis, as well as to evaluate the efficacy of a combined antimicrobial treatment with glucose in eradicating S. sanguinis persisters infections. Collectively, these findings may contribute to the formulation of novel therapeutic strategies for infective endocarditis (IE) resulting from oral streptococcal infections.
I believe the manuscript is logically structured and holds significant potential for publication, provided that the following issues are addressed: 1. Are there alternative methods to eliminate persister cells? It appears that 10 mg/ml glucose and 100 mg/ml glucose yield similar results, with 10 mg/ml glucose potentially being slightly more effective. What accounts for this observation? This section could benefit from an expanded discussion. 2. I observed discrepancies in the results obtained using PCG/GM as shown in Fig. 3 compared to those presented in Figs. 4 and 5; specifically, the number of surviving cells after 24 hours differs by two orders of magnitude. What explains this variation? 3. In terms of practical significance, it would be beneficial to enhance the literature by exploring current countermeasures against persisters and highlighting any advancements made in clinical settings.
Author Response
Responses to the comments from Reviewer 1
Thank you for taking the time to review our manuscript. Please find the detailed responses below and the corresponding revisions and corrections yellow-highlighted in the re-submitted files.
Comment 1: Are there alternative methods to eliminate persister cells?
Response 1: Thank you for your question. Previous reports have shown that anticancer agents such as cisplatin and mitomycin C are effective in killing E. coli persisters [35,36]. However, there are concerns about the high burden of these drugs on the human body due to their potent effects. Therefore, we sought a substance with a low impact on the human body and, thus, focused on glucose. We have added this explanation to the Discussion section (lines 198–202).
[35] doi:10.1002/bit.25963, [36] doi:10.1111/1462-2920.12873
Comment 2: It appears that 10 mg/ml glucose and 100 mg/ml glucose yield similar results, with 10 mg/ml glucose potentially being slightly more effective. What accounts for this observation? This section could benefit from an expanded discussion.
Response 2: Thank you for pointing this out. The difference in effect between 10 mg/mL glucose and 100 mg/mL glucose was small and both were completely sterilized after 24 hours. We consider that increasing the amount of added glucose from 10 mg/mL to 100 mg/mL did not change the bactericidal effect of the antimicrobials. Therefore, we consider this difference to be an error due to the experimental technique.
Comment 3: I observed discrepancies in the results obtained using PCG/GM as shown in Fig. 3 compared to those presented in Figs. 4 and 5; specifically, the number of surviving cells after 24 hours differs by two orders of magnitude. What explains this variation?
Response 3: Thank you for pointing this out. In Figure 3, BHIY broth was used as the culture medium, while in Figures 4 and 5, M9 minimal medium was used as the culture medium. This may explain the difference in the number of viable bacteria after 24 hours. Additionally, in the Materials and Methods, “BHIY” was incorrect in lines 303 and 305 and was corrected to “M9 minimal medium”.
Comment 4: In terms of practical significance, it would be beneficial to enhance the literature by exploring current countermeasures against persisters and highlighting any advancements made in clinical settings.
Response 4: Thank you for your comment. We agree with you. Since this paper only shows in vitro results from basic research, as you suggested, future work should examine the effects in clinical settings. This is included in the “Study limitations” section in the Discussion.
Reviewer 2 Report
Comments and Suggestions for Authors
The study investigated the survival of S. sanguinis persisters following treatment with Penicillin and Gentamicin. The study identified effective treatment strategies against persisters using glucose. To emphasize the complete nature of the study, authors should consider changing the title to read “Glucose Supplementation Enhances the Bactericidal Effect of Penicillin and Gentamicin on Streptococcus sanguinis Persisters”
The article is better classified as a “short communication” or simply “communication”
Line 88: Since Figure 1 shows the MIC results, please delete “Determination of the” from the title.
Line 89: Please clarify if brain heart infusion agar was used or BROTH. Broth should be more appropriate here.
Line 119: Please change the title of Figure 3 to “Persister Formation in S. sanguinis Following PCG/GM Treatment and Recovery in BHIY Medium”
Line 139: Also, change the title of Figure 4 to “Bactericidal effect of PCG (1600 μg/mL)/GM (64 μg/mL) on S. sanguinis in M9 minimal medium following the addition of glucose (1, 10, or 100 mg/mL).”
Line 206: There is no information about the source of S. sanguinis ATCC 10556 and why the strain was selected for this study. If it was obtained directly from ATCC, it should be stated and when. Is it a clinical strain?
Line 220: Absorbance of 0.05 at 600 nm corresponds to which CFU/ML? Why this absorbance level?
Moreover, CONTROLS for MIC determination, bactericidal effect tests, and glucose experiments should be stated in their respective sections in the Materials and Methods.
Author Response
Response to the comments from Reviewer 2
Thank you for taking the time to review our manuscript. Please find the detailed responses below and the corresponding revisions and corrections yellow-highlighted in the re-submitted files.
Comment 1: To emphasize the complete nature of the study, authors should consider changing the title to read “Glucose Supplementation Enhances the Bactericidal Effect of Penicillin and Gentamicin on Streptococcus sanguinis Persisters”
Response 1: Thank you for your very important point. As you suggested, we revised the title.
Comment 2: The article is better classified as a “short communication” or simply “communication”
Response 2: Thank you for your suggestion. The results of the arginine effect have been added to Figures 4 and 5. Note that we also observed the bactericidal effect of arginine on persisters, but this effect was higher with glucose. We consider that our study should be submitted as an “Article” rather than a “Communication” because this adds depth to the content.
Comment 3: Line 88: Since Figure 1 shows the MIC results, please delete “Determination of the” from the title.
Response 3: Thank you for pointing this out. “Determination of the” has been deleted in line 94.
Comment 4: Line 89: Please clarify if brain heart infusion agar was used or BROTH. Broth should be more appropriate here.
Response 4: Thank you for pointing this out. The word “agar” has been corrected to “broth” in line 95.
Comment 5: Line 119: Please change the title of Figure 3 to “Persister Formation in S. sanguinis Following PCG/GM Treatment and Recovery in BHIY Medium”
Response 5: Thank you for your suggestion. The title of Figure 3 has been changed (line 126).
Comment 6: Line 139: Also, change the title of Figure 4 to “Bactericidal effect of PCG (1600 μg/mL)/GM (64 μg/mL) on S. sanguinis in M9 minimal medium following the addition of glucose (1, 10, or 100 mg/mL).”
Response 6: Thank you for your suggestion. The title of Figure 4 has been changed (line 151).
Comment 7: Line 206: There is no information about the source of S. sanguinis ATCC 10556 and why the strain was selected for this study. If it was obtained directly from ATCC, it should be stated and when. Is it a clinical strain?
Response 7: Thank you for pointing this out. S. sanguinis ATCC10556 is held by ATCC as a strain isolated from a patient with subacute bacterial endocarditis [40]. We therefore obtained this strain from ATCC and used it as our target bacteria. We added this information in lines 249–251.
[40] doi: 10.1099/00207713-39-4-471
Comment 8: Line 220: Absorbance of 0.05 at 600 nm corresponds to which CFU/ML? Why this absorbance level?
Response 8: Thank you for your question. The absorbance of 0.05 when measured at 600 nm was determined based on a previous study [25]. Therefore, we do not know the specific number of bacteria. However, since the MIC measurement was performed with sufficient dilution, we do not consider this to be a problematic experimental method.
Comment 9: CONTROLS for MIC determination, bactericidal effect tests, and glucose experiments should be stated in their respective sections in the Materials and Methods.
Response 9: Thank you for pointing this out. As suggested by you, we have added a description of the controls in lines 268, 278, and 309.
Reviewer 3 Report
Comments and Suggestions for Authors
Main Question
Why did the authors choose to conduct relatively simple, inexpensive, and quick experiments using only one streptococcal strain? This choice results in a minimal amount of data (two antibiotics plus glucose) obtained from a single strain, as seen in the "Results" section, which translates into a weak "Discussion" section. The discussion is perhaps the weakest part of the work and requires expansion, reorganization, and enrichment with content related to potential mechanisms explaining the presented results.
Introduction
I believe more information is needed to understand the role of persister cells fully. For example:
Within the population of a given species, do persister cells differ genetically from other sensitive cells?
What is their survival mechanism if they do not exhibit resistance?
More details are needed to clarify this part.
Figures
The paper contains numerous figures, but they are very simple and provide limited information. It might be worth considering whether they could be combined in some way. This is particularly noticeable in Figure 5, which presents a very small amount of data.
Results
The results are very modest and scarce. They are supplemented with figures, but the experiments remain very basic (valuable, but limited in quantity and presentation).
Discussion
The beginning of the discussion essentially repeats the introduction. This should be rephrased to avoid redundancy. Specific points that require attention include:
What is the role of glucose in the PCG/GM combination therapy? Is it used in practice? If so, how? Provide more details.
Specific Comments
Line 90: "Different concentrations of PCG and GM were incubated for 24 h" — Please clarify that this refers to media supplemented with specific concentrations of antibiotics.
Line 101: Specify which guidelines are being referenced.
Line 107: Does the statement "the standard deviation across at least three biological replicates" imply varying replication numbers for different time points?
Lines 182–184: How is this expected to occur? Are we discussing proton motive force in relation to bacterial cells? Glucose does not directly generate proton motive force but is oxidized to provide substrates for the respiratory chain. How is this linked to antibiotic uptake?
Line 200: Does using higher-than-physiological concentrations of glucose during therapy truly lead to type II diabetes? How long does such treatment last? Is this statement accurate?
Lines 221–224: Was homogeneity of samples ensured during this procedure? I have often encountered differing measurement results for bacteria in small volumes on microplates analyzed with a microplate reader compared to measurements in cuvettes tested on standard spectrophotometers.
References
A large portion of the references are quite outdated (e.g., from 1962, 1975, 2001, 1942, 2000). It may be worthwhile to seek more recent sources.
Author Response
Response to the comments from Reviewer 3
Thank you for taking the time to review our manuscript. Please find the detailed responses below and the corresponding revisions and corrections yellow-highlighted in the re-submitted files.
Comment 1: Why did the authors choose to conduct relatively simple, inexpensive, and quick experiments using only one streptococcal strain? The discussion is perhaps the weakest part of the work and requires expansion, reorganization, and enrichment with content related to potential mechanisms explaining the presented results.
Response 1: Thank you for your helpful comments. In this study, we focused our experiments on infective endocarditis. S. sanguinisATCC10556 was used in this study because it was isolated from a patient with subacute bacterial endocarditis (line 249). Since the role ofS. sanguinis persister cells in infective endocarditis had not been reported yet, to the best of our knowledge, we first conducted experiments using a single strain. Although we chose a simple experimental method for this study, we believe that we have obtained important results for future medical contributions in proving the identification of S. sanguinis persister cells and the method for their eradication. We have also added the arginine effects in Figures 4 and 5. Note that we also observed the bactericidal effect of arginine on persisters, but this effect was higher with glucose. We have also included a discussion of these results, which adds depth to the discussion section. As you pointed out, the potential mechanisms have not been elucidated and need to be clarified in the future (this is mentioned in the “Study limitations” section).
Comment 2: I believe more information is needed to understand the role of persister cells fully. For example: Within the population of a given species, do persister cells differ genetically from other sensitive cells? What is their survival mechanism if they do not exhibit resistance?
Response 2: Thank you for pointing this out. Persister cells are not genetically different from other sensitive cells. The survival mechanism is known to be a nongrowing state and survival under stressful conditions. This information is included in the description of persisters in lines 58 and 60. The importance of the toxin-antitoxin system in persister formation has also been partly added in the Introduction. However, since this study is mainly focused on clarifying and eliminating the existence of persisters rather than elucidating the mechanism, we have chosen to keep the information and explanation on the mechanism of persisters simple.
Comment 3: The paper contains numerous figures, but they are very simple and provide limited information. It might be worth considering whether they could be combined in some way. This is particularly noticeable in Figure 5, which presents a very small amount of data. The results are very modest and scarce. They are supplemented with figures, but the experiments remain very basic (valuable, but limited in quantity and presentation).
Response 3: Thank you for pointing this out. The previous data were indeed not informative, but now that we have added the new arginine-verified data, there is more information available from the figure than before. Accordingly, we have provided the necessary information in the results.
Comment 4: The beginning of the discussion essentially repeats the introduction. This should be rephrased to avoid redundancy. Specific points that require attention include: What is the role of glucose in the PCG/GM combination therapy? Is it used in practice? If so, how? Provide more details.
Response 4: Thank you for pointing this out. We have rephrased lines 178–182 to avoid redundancy. The role of glucose is to enhance the uptake of an antimicrobial agent by increasing the proton motive force. It is not used in practice.
Comment 5: Line 90: "Different concentrations of PCG and GM were incubated for 24 h" — Please clarify that this refers to media supplemented with specific concentrations of antibiotics.
Response 5: Thank you for pointing this out. “BHIY broth included” has been added in line 95.
Comment 6: Line 101: Specify which guidelines are being referenced.
Response 6: Thank you for pointing this out. “AHA” has been added in line 107.
Comment 7: Line 107: Does the statement "the standard deviation across at least three biological replicates" imply varying replication numbers for different time points?
Response 7: Thank you for pointing this out. "Experiments were performed in cultures from different single colonies" has been added in line 113.
Comment 8: Lines 182–184: How is this expected to occur? Are we discussing proton motive force in relation to bacterial cells? Glucose does not directly generate proton motive force but is oxidized to provide substrates for the respiratory chain. How is this linked to antibiotic uptake?
Response 8: Thank you for pointing this out. Though the complete mechanism of aminoglycoside uptake is unclear, it is known that a threshold PMF is required. Glucose is transported to the cytoplasm and enters glycolysis, where its catabolism generates nicotinamide adenine dinucleotide. It is oxidized by enzymes in the electron transport chain that contribute to PMF. We consider that elevated PMF enhances aminoglycoside uptake and the bactericidal effect [29]. We have added this explanation to the Discussion section (lines 209–214).
Comment 9: Line 200: Does using higher-than-physiological concentrations of glucose during therapy truly lead to type II diabetes? How long does such treatment last? Is this statement accurate?
Response 9: Thank you for pointing this out. The risk of type 2 diabetes is listed [38] because this study was not performed in humans. Treatment lasts 2–6 weeks [27]. We have added this information to the Discussion section (lines 238–240).
[38] doi:10.3390/ijms21176275
Comment 10: Lines 221–224: Was homogeneity of samples ensured during this procedure? I have often encountered differing measurement results for bacteria in small volumes on microplates analyzed with a microplate reader compared to measurements in cuvettes tested on standard spectrophotometers.
Response 10: Thank you for your input. In this study, we did not compare absorbance measurements performed with a microplate reader with those that used a cuvette, so we do not know if there are differences between them, but we believe that we should focus only on the absorbance of each sample in the plate reader since the MIC measurement confirms the final growth point by measuring absorbance in the plate reader for all samples.
Comment 11: A large portion of the references are quite outdated (e.g., from 1962, 1975, 2001, 1942, 2000). It may be worthwhile to seek more recent sources.
Response 11: Thank you for pointing this out. Following your suggestion, references Duperval, R. et al. 1975 (previously [31]) and Levy, C.S. et al. 2001 (previously [32]) have been changed to Pericas, J.M. et al. 2019 (currently [32]) and Matsuo, T. et al. 2021 (currently [33]). Other references have not been changed as we believe they are best suited for the relevant citation (for example, the paper on the discovery of persisters: [13] Hobby, G.L, et al. 1942).
Reviewer 4 Report
Comments and Suggestions for Authors
In this manuscript by Takada et al., the authors identified the presence of persister cells in Streptococcus sanguinis in vitro, the primary organism causing infective endocarditis (IE). The manuscript also showed that with the addition of glucose, the persister population can be limited or completely eliminated by the combination treatment of PCG/GM.
Overall, this manuscript is well organized, and the data presented support the conclusion in general.
Comments that may further strengthen this manuscript are provided as follows:
1. A possible interpretation for the growth curve of 1mg/ml group is that persister cells resuscitated in fresh M9 with glucose and some of them are killed by antibiotics. But since glucose was used up before all cells were killed by antibiotics, surviving cells become persistent again. So, if the glucose can be maintained at the same level, would all cells be killed eventually even with lower concentration of glucose? Please comment.
2. Would Streptococcus sanguinis cells in blood vessels ever experience nutrient starvation? If not, the recurrence of IE may not be attributed to persister cells.
3. Would the bactericidal effect of individual antibiotic also be enhanced in the presence of glucose?
4. Could a similar effect be achieved by adding other types of carbon source, as previously explored in reference 28? If yes, would any of these alternatives be more suitable for clinical use than glucose?
5. Would providing other types of nutrient (such as amino acids) in addition to glucose further enhance the bactericidal effect?
Author Response
Response to the comments from Reviewer 4
Thank you for taking the time to review our manuscript. Please find the detailed responses below and the corresponding revisions and corrections yellow-highlighted in the re-submitted files.
Comment 1: A possible interpretation for the growth curve of 1mg/ml group is that persister cells resuscitated in fresh M9 with glucose and some of them are killed by antibiotics. But since glucose was used up before all cells were killed by antibiotics, surviving cells become persistent again. So, if the glucose can be maintained at the same level, would all cells be killed eventually even with lower concentration of glucose? Please comment.
Response 1: Thank you for your helpful comments. As you pointed out, if glucose, PCG, and GM can be maintained at the same concentration, persister cells will be eventually killed even if the glucose concentration is low. However, we speculate that glucose resuscitated the persister cells and that the effect of the antibiotic was exerted by this. As mentioned in the “Study limitations” section, it is necessary to elucidate the detailed mechanism by which the addition of glucose improved the effect. If this hypothesis is substantiated, we believe that methods to maintain concentrations, even if those concentrations are low, will eventually kill all bacteria, including persister cells, as suggested by you.
Comment 2: Would Streptococcus sanguinis cells in blood vessels ever experience nutrient starvation? If not, the recurrence of IE may not be attributed to persister cells.
Response 2: Thank you for pointing this out. Since the blood always contains sugars and amino acids, it is unlikely that S. sanguinis will starve in the blood vessels. However, starvation is not the only factor that triggers the formation of persister cells. Stress from antimicrobial agents alone can also trigger persister cell formation, as mentioned in the Introduction (lines 55 and 56).
Comment 3: Would the bactericidal effect of individual antibiotic also be enhanced in the presence of glucose?
Response 3: Thank you for pointing this out. In this study, we did not measure the potentiating effect of glucose on individual antibiotics because the experiments were specifically focused on antibiotic use according to the guidelines (i.e., PCG/GM combination). However, as you noted, the bactericidal effect of individual antibiotics may be enhanced in the presence of glucose. In particular, since the combination of glucose and aminoglycoside antimicrobials has been demonstrated by Allison, K.R. et al. to be effective in killing persister cells [29], it is highly likely that glucose will enhance gentamicin alone.
Comment 4: Could a similar effect be achieved by adding other types of carbon source, as previously explored in reference 28? If yes, would any of these alternatives be more suitable for clinical use than glucose?
Response 4: As you pointed out, other types of carbon sources could have the same effect. S. sanguinis can also metabolize other sugars such as sucrose [39], so other sugars may need to be investigated as carbon sources. We believe that glucose is more suitable than other carbon sources for clinical use because its blood levels are easier to measure.
[39] doi:10.1111/apm.12238
Comment 5: Would providing other types of nutrient (such as amino acids) in addition to glucose further enhance the bactericidal effect?
Response 5: Thank you for your question. As S. sanguinis can metabolize arginine, an amino acid, additional experiments were conducted on enhancing the antibiotic's bactericidal effect with arginine. The results showed that arginine also enhanced the effect, although not as much as glucose. These results have been added to Figures 4 and 5. Hence, we believe that adding other nutrients, such as amino acids, along with glucose may enhance the bactericidal effect. The combination of glucose and other nutrients is one of the prospects for future research.
Round 2
Reviewer 3 Report
Comments and Suggestions for Authors
The revised version of the work is significantly improved. I have received satisfactory responses to the vast majority of my concerns and comments.
In my opinion, the paper can be accepted.